# Sustainable Survival: Resource Management Strategy in Micro and Small Enterprises in the Rubber Products Market in Poland during the COVID-19 Pandemic

Katarzyna Czainska [1,*] , Aleksandra Sus [1] and Eleftherios I. Thalassinos [2,3,*]

1   General Tadeusz Kościuszko Military University of Land Forces, 51-147 Wrocław, Poland;
    aleksandra.sus@awl.edu.pl
2   Faculty of Maritime and Industrial Studies, University of Piraeus, 185-33 Piraeus, Greece
3   Faculty of Economics, Management and Accountancy, University of Malta, 2080 Msida, Malta
*   Correspondence: katarzyna.czainska@awl.edu.pl (K.C.); thalassinos@ersj.eu (E.I.T.)

**Abstract:** The COVID-19 epidemic surprised economic operators around the world. The very existence of many businesses, and thus jobs, was at stake. However, one year after the WHO declared the outbreak a pandemic, contrary to the pessimistic forecasts of business analysts, some industries did not experience the predicted negative effects of the crisis. This article presents the results of a pilot study on micro and small enterprises in the rubber products industry in Poland during the COVID-19 pandemic, with the aim of analyzing the phenomenon of sustainable resource management that led not only to the survival of these enterprises but also to a significant increase in their turnover. Therefore, the purpose of the study was to analyze the key success factors of the indicated economic entities, with particular emphasis on the perspective of sustainable resource management and relationship management. On the basis of best research practices, a triangulation of research methods was applied (integrative literature review, computer-assisted telephone interviewing, and individual in-depth interview). A relationship was observed between the sustainable management of resources and the structure of the relationship network and the strength of its connections. In micro and small enterprises in the rubber products sector in Poland, sustainable resource management is related to the structure of the network of relations and the strength of connections in the network (relations/networking), as enterprises form a group of entities with a high level of loyalty, especially between the suppliers and buyers of raw materials. The formulated conclusions will become the basis for further in-depth research that can be conducted (a) in the same group of respondents, but using a representative research group, (b) in the same industry among a group of large enterprises, and (c) in a group of small and medium-sized enterprises (SMEs) from other industries.

**Keywords:** sustainability; relations; resource management; rubber products market; SME; COVID-19

## 1. Introduction

Since 2007, the Global Risks reports published by the World Economic Forum [1] have identified pandemics at the forefront of potential threats in terms of their high probability of occurrence, large scale, and significant impact. On 11 March 2020, one of the predicted negative scenarios was realized. Director-General of the World Health Organization (WHO), Tedros Adhanom Ghebreyesus, stated that the COVID-19 disease caused by the SARS-CoV-2 coronavirus could be classified as a pandemic, i.e., according to the adopted definition of the worldwide spread of a new disease [2]. The Organisation for Economic Co-operation and Development (OECD) analysts promptly identified at-risk industries and estimated a looming decline in production in individual sectors. Their research predicted an activity decrease of (a) 100% for companies related to the production of transport equipment, real estate services, activities related to culture, entertainment, and recreation, and other service activities; (b) 75% for wholesale and retail trade companies, air

transport, hospitality, and catering services; and (c) 50% for construction and professional services companies. A 15% drop in revenue was expected for the remaining non-financial sectors [3]. In terms of the size of enterprises, the small and medium enterprise (SME) sector was characterized as vulnerable to the crisis. A survey conducted at the end of June 2020 showed that almost 90% of small businesses had experienced a strong (51%) or moderate (38%) negative impact of the pandemic, 45% of companies had experienced supply chain disruptions, and 25% of companies had less than 1–2 months' cash reserves [4] (p. 5). At the end of September 2020, Statistics Poland estimated that there were 4.1 million registered economic activities in Poland, of which 2.6 million were actively operating entities [5]. SMEs played a key role in this ranking. According to the data for 2019, there were 2.15 million enterprises in Poland, of which (according to the number of employees) 99.8% were identified as SMEs (96.7% micro enterprises; 2.4% small enterprises). The SME sector generates 49.1% of Poland's GDP, and micro enterprises make the largest contribution at about 30.3% [6]. The presented structure of the Polish market, with the overwhelming number of entities from the SME group, could therefore raise considerable concern in the face of a pandemic threat. Thus, there was a real threat to Polish micro and small enterprises specializing in the production of rubber products. This industry, however, resisted the crisis, prompting business analysts to try to explain this phenomenon.

The rubber products market is a subsystem of the plastics and rubber products market, which in turn is a part of the chemical market. According to the Polish Classification of Activities (PKD 2007) [7], the rubber products market is code 22.1. This corresponds to the NACE (Statistical Classification of Economic Activities in the European Community) [8]. Economic activity in the field of plastics and rubber is carried out by a total of 15,068 enterprises, of which 8984 are natural persons running a business [4]. This article focuses on the production of other rubber products (class 22.19) carried out by Polish micro and small enterprises. The target group of surveyed entities was selected due to: (1) the significant importance of the Polish chemical industry for the European market, (2) the significant importance of micro and small companies for the Polish economy, and (3) the stable development of this sector, even during the COVID-19 pandemic. The Polish chemical industry, which includes the rubber industry, is the leader of chemical industries in Central and Eastern Europe, creating 327,000 jobs and accounting for 12% of total employment [9]. In 2019, the sold production of industrial products of the chemical industry in Poland amounted to PLN 41.74 billion, of which the plastic and rubber products industry accounted for PLN 13.11 billion [10]. The impact of the COVID-19 pandemic on the economic situation was thus much greater than originally forecast. It should be noted that the rubber and plastic products sub-sector recorded the largest decrease in average employment in January–September 2020, decreasing by an average of 6000 people, accompanied by the largest incremental increase in sold production. It should also be noted that the industry in question was not fully covered by the Polish government support program called the "Anti-Crisis Shield" [11,12].

The above facts prompted the formulation of research questions that generally relate to sustainable resource management in the event of a threat to the functioning of an enterprise with small capital and market share. The selected companies are a tightly knit group of market players that did not suffer from the economic crisis in 2008–2009 but were much less able to defend themselves against the effects of the global COVID-19 pandemic than the bulk chemicals and fuel sectors. Additionally, the research area becomes even more interesting once the assessment includes the current operating conditions (i.e., the drastic increase in the price of transport from China recorded at the end of March 2021). The identification of further key factors is the subject of this research, whose results are presented in this article. The results of the presented study are of key importance in shaping the crisis prevention policy. Conclusions drawn from the analysis of sectors that have survived the threat will contribute to the development of good practices for other sectors. Most of the identified solutions should be implemented while the environment is stable.

## 2. From Sustainable Development to Sustainable Resource Management

In reference to the formulated research questions, the literature was reviewed with a focus on the issue of sustainable resource management in crisis conditions. Particular attention was paid to works on micro and small enterprises. Of course, first, it was necessary to verify that the COVID-19 pandemic could be characterized as a crisis situation.

From the social and economic points of view, the COVID-19 pandemic meets all of the criteria for a crisis situation on a global scale. The crisis caused by the COVID-19 pandemic does not involve only one aspect of the functioning of societies. Instead, it affects many spheres of activity, and thus, we can classify it into all major types of crises, namely: poverty-related, unemployment, economic, financial, environmental, international, informational, physical, human resources, and reputational [13]. In economic activity, a crisis is defined as an inherently abnormal, complex, and unstable situation that represents a threat to the strategic objectives, existence, or reputation of the organization [14]. It is also noted that a crisis in business is "a low-probability, high-impact situation that is perceived by critical stakeholders to threaten the viability of the organization" [15].

The main threats to micro and small enterprises resulting from the crisis caused by the pandemic include: loss of revenue, reduction in sales, loss of employment, loss of financial liquidity, increase in debt, disruption of supply chains, and even the liquidation of the enterprise [13,16–18]. In the case of COVID-19, the literature on the subject formulates two terms that collectively define the changes that have occurred: "demand shock" and "supply shock". Supply shock was influenced by the closure of some factories and the disruption of the supply chain. The demand shock, in turn, was influenced by the closure of shops and warehouses [13,19,20].

Therefore, since we are dealing with a global crisis, we must use extensive systemic solutions to overcome it. The basis of these solutions is the guidelines related to crisis management and sustainable resource management (SRM).

As crisis management (CM) encompasses many spheres of impact (psychological, cognitive, communication, financial, safety, health, etc.) [13,21], the range of recommended solutions is also wide. Based on research conducted during the COVID-19 pandemic, the following actions have been recommended as effective strategies: reaching new market segments; extending marketing and advertising campaigns; offering high discounts; providing and promoting special offers and prices; lowering the prices of products and services; researching and understanding the needs of target customers and the changes taking place; focusing on loyal customers; increasing the marketing budget; reducing employees' salaries, providing paid vacation and compulsory leave, dismissing employees to reduce the workforce, increasing the scope of staff duties, reducing staff working time, and replacing permanent employees with part-time temporary employees; deferring certain payments and/or rescheduling payments; developing additional revenue opportunities; closing some departments; using new technologies to reduce operating costs; reducing the budget of investments aimed at expansion; and cooperating with other entities and organizing meetings to discuss ways to overcome the crisis [16,18,20,22,23]. Therefore, addressing this issue involves a specific mixture of solutions that can be defined as a combination of short- and long-term strategies, including future development aspirations [24].

The above recommendations, however, prompt reflection on the discrepancy between management in a crisis and management in the era of stability, i.e., between management focused on survival and sustainable management/development.

Due to the holistic nature of the objectives formulated in the guidelines specified in the UN document entitled "Transforming our world: the 2030 Agenda for Sustainable Development" [25], it can be concluded that sustainable management is a complex concept and cannot be reduced, for example, to the ecological aspect [26,27], which is often the case in the literature on the subject [28]. Instead, it is prudent to adopt a perspective in which sustainably addresses the interests of stakeholders [26,29–34]. The presented view is dominated by the following approaches: systemic (organization as a system functioning in, for, and due to the environment), strategic (multi-faceted planning and focused on achiev-

ing results over a longer period of time, taking into account the interests of stakeholders), and resource-focused (rationalizing the acquisition and use of resources). Therefore, the following assumptions are proposed:

- A sustainable organization is one that achieves objectives that ensure, depending on the adopted strategy, its stability or growth using methods and tools for achieving strategic objectives that are accepted by key external and internal stakeholders in terms of social justice and environmental safety;
- A sustainable enterprise is one that achieves economic results ensuring, depending on the adopted strategy, its stability or growth, while methods and tools for achieving strategic goals are accepted by key external and internal stakeholders in terms of social justice and environmental safety [29].

Consequently, sustainable resource management (SRM) should be understood as activities of a strategic nature, the basis of which is the effective protection of resources necessary for, depending on the adopted strategy, the stability or growth of the organization, while the methods and tools for achieving strategic goals are accepted by key stakeholders, both external and internal, in terms of social justice and environmental safety. The aspects of SRM can also be supplemented with the concept of "maturity of the responsible resource management", i.e., practices defined as the extent to which resources are identified, controlled, and optimized with regards to their responsible management [35].

Taking the stakeholder perspective changes the context of sustainable management and development. This forces management theorists and practitioners to return to activities known from strategic management, namely, identifying stakeholders (including key entities), defining their interests and needs, and developing strategies that enable the balancing of interests and needs [19,36,37].

The immediate consequence of adopting the stakeholder perspective paradigm is the need to focus on the significant importance of building relationships and, as emphasized above, basing crisis management on maintaining old and creating new relationships. Relations in strategic terms usually take the form of cooperation, competition, and coopetition [38,39]. Cooperation refers to relations with environmental elements that make it possible to generate common benefits in the form of increased efficiency of undertaken activities, easier achievement of goals, and minimization of selfish behaviors. This type of relationship increases the probability that new value will emerge from taking advantage of opportunities. It is also associated with lower risks and costs of launching new or improved products in the market, an increase in the time to market, and easier access to new markets [39,40]. It is emphasized that thanks to new organizational solutions, self-organization processes are also carried out in the implementation of joint business projects. This requires knowledge of the mechanisms that allow for the identification and understanding of opportunities, including organizational opportunities. In the classic approach, competitive relations are formed between the organization and elements of the competitive environment, perceived in terms of products and markets. Porter's approach, however, does not reflect the complexity and multidimensionality of the discussed relations in the contemporary approach. The dynamic perception of relations between competitors mainly concerns resources and competencies resulting from the increased concentration of enterprise activities in market niches, cooperation, and efforts to achieve uniqueness. Such behavior shortens the life cycle of opportunities while at the same time triggering aggressive competition for new development opportunities that do not yet exist. In this way, a new view of competition emerges—competing for market opportunities—at least in the theoretical field. In practice, a great deal of evidence suggests that product cannibalism causes other possibilities for action to be overlooked (disruptive technologies) [41–44].

In a crisis situation, coopetition, i.e., a relationship based on simultaneous cooperation and competition, may prove useful, as it is designed to achieve higher operational efficiency while allowing all participating entities to benefit in the process [45–47]. Thus, it is a positive-sum game, although the literature also emphasizes its neutral and negative roles [48,49]. The main premises of coopetition are the benefits resulting from the inno-

vative development and market development of companies. This means that companies strive to reduce uncertainty and risk by jointly developing new solutions, including in terms of products and markets. The coopetition strategy offers benefits to all parties involved in its implementation (partners), not only to selected entities, as is the case with pure competition or cooperation. Therefore, this solution appears to be a necessary priority for modern enterprises struggling with pandemic problems.

The third pillar of survival in the age of a pandemic (apart from sustainable survival and relations/networking) is diversification. It has been shown that both product and geographic diversification has helped companies in various sectors survive in the pandemic. Moreover, it is thanks to diversification that companies are better able to survive the COVID-19 pandemic than they were in the 2008 financial crisis [50]. However, there is no consensus as to the preferred form of diversification, as the choice between the concentric and conglomerate diversification strategy depends on many market and intra-organizational factors. With the above considerations in mind, diversification can be considered a form of sustainable resource management and a distributed network of relationships.

## 3. Materials and Methods

Empirical research was preceded by an integrative literature review. The aim of the study was to identify the predicted business threats to micro and small enterprises during the COVID-19 pandemic and the crisis management solutions used in this group. Leading databases (Web of Science, Scopus, EBSCO, ResearchGate, Google Scholar) were queried to find articles published in a specific time period (publications from 2020 to 2021) and exploring the subject ("COVID-19"; "crisis"; "SME"); as a result, 9007 articles were obtained. Then, the results were filtered by discipline (publications in the fields of economics, business, and management were selected). The number of publications was thus reduced to 463. Then, articles on micro and small businesses were selected, narrowing the results down to 18 items. The literature review was supplemented with items that do not meet the above-mentioned criteria but are necessary to explain the discussed theoretical concepts.

As a result of the literature review, a research gap was identified, the area of which can be described as: sustainable resource management in micro and small enterprises in the rubber products industry in Poland during the COVID-19 pandemic. In order to clarify the subject of the research, the following research questions were asked:

1. What are the most important resources of micro and small enterprises in the rubber products industry in Poland?
2. What activities do micro and small enterprises in the rubber products industry in Poland associate with "sustainable resource management"?
3. What activities in the field of "sustainable resource management" are undertaken by micro and small enterprises in the rubber products industry in Poland?
4. What activities in the field of "sustainable resource management" were taken by micro and small enterprises in the rubber products industry in Poland during the COVID-19 pandemic?
5. What activities did micro and small enterprises in the rubber products industry in Poland take during the COVID-19 pandemic to maintain their operations?

Three research theses were also formulated:

- (T1) In micro and small enterprises in the rubber products sector in Poland, sustainable resource management is related to the structure of the relationship network and the strength of connections in this network (relations/networking).
- (T2) When the survival of micro and small enterprises in the rubber products sector in Poland is threatened, concentric diversification becomes the foundation of sustainable resource management.

- (T3) In the event of a threat to the survival of micro and small enterprises in the rubber products sector in Poland, sustainable resource management consists in securing the interests of key stakeholders in the existing relationship network (sustainable survival).

Empirical (qualitative and quantitative) research was carried out in the period March–May 2021 using the following methods:

a. Computer-assisted telephone interviewing (CATI) of 19 micro and small enterprises in the rubber products sector in Poland: the questionnaire was sent to 50 randomly selected enterprises that met the following criteria: (a) they were entered in the Polish National Court Register under PKD code 22.19; (b) they were classified as micro or small enterprises (by the number of employees) according to EU recommendations [51]. Only responses that addressed all questions were included in the analysis. The data obtained from 19 respondents were analyzed.

b. Individual in-depth interview (IDI) with owners of micro and small enterprises in the rubber products sector in Poland: 5 respondents were interviewed (candidates were selected from among people who declared their willingness to participate in the interview during CATI and were owners of enterprises).

The authors of the article conducted the interviews as part of the CATI and IDI surveys.

With the CATI method, an original questionnaire was developed (Appendix A). In the in-depth interview (Figure 1), the respondent was allowed full freedom of expression, and the interviewer directed the interview in such a way that the issues formulated in the research questions were raised. It is important to note that the respondents were asked to answer the questions only from the perspective of their own enterprise.

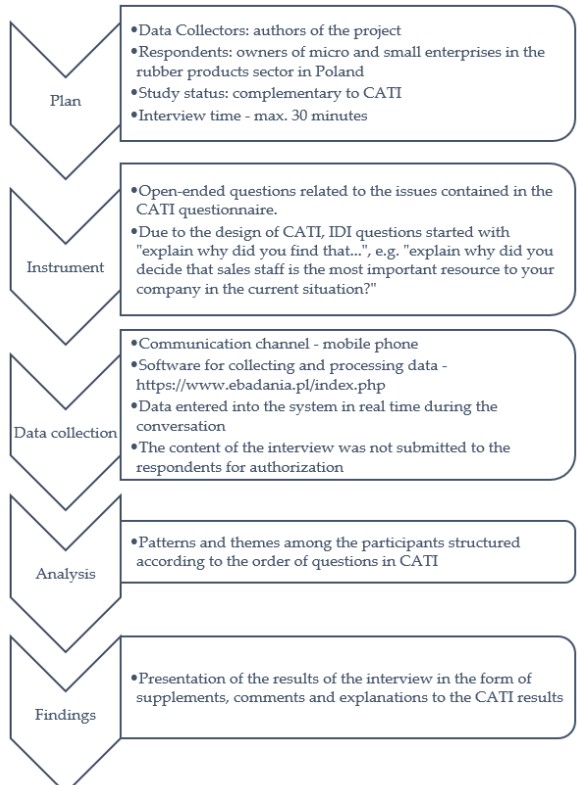

**Figure 1.** The process for conducting in-depth interviews.

## 4. Results

Among the 19 enterprises whose responses were taken into account in the survey, small enterprises (12) were the majority, while the remaining ones (7) were micro enterprises. Most of the surveyed companies had been operating for over 20 years (11 companies), and

2 companies had been on the market for less than 3 years. The interviewees were business owners (63%), managers (26%), and employees (11%).

The first question concerned the identification of key resources for the functioning of the enterprise during the pandemic. The answers were classified into four groups: human, financial, material, and information resources. In the human resources group, production employees (84% of responses) and sales (53%) were indicated as key. The structure of the answers did not differ on the basis of the applied criteria of the respondents, i.e., the size of the company, years on the market, or the position of the respondent in the organization. In the financial resources group, the most important factors for the respondents were the funds in the company's bank account (89% of responses), cash (26%), and receivables (37%). Entrepreneurs believed that only the company's own funds were a guarantee of survival; external funds, e.g., those offered as part of government support (5%) or loans/credits (11%), were not an attractive form of securing the business. Among physical resources, the most important were raw materials (74%) and machinery and equipment (58%). Due to the business profile of the surveyed enterprises (manufacturing industry), the given responses only confirm the strategic rationality of the owners/respondents. Relatively little importance was attached to the means of transport (16%), as these companies use logistics services offered by external suppliers. The interviewees most commonly referenced UPS, DPD, FedEx, Schenker, and DHL Parcel as logistics companies with which they have signed long-term contracts. This is due to the fact that the surveyed SMEs most often serve diverse customers located in various places in Poland, and their products are not mass-produced. The slow recovery of wholesale and large-scale orders can be observed, mainly due to rising shipping prices in China. However, the process will be evolutionary rather than revolutionary. Moreover, Polish companies from the rubber industry are used to manufacturing short series of non-standard products, i.e., those that require direct contact and individual arrangements between the manufacturer and the supplier (purchase order). Such activities are unprofitable and often ineffective for Chinese producers due to the lack of full control over the final product, delivery time and price, and geographical remoteness.

In the information resources group, the distribution of responses was almost uniform, as the respondents indicated the following elements as important: industry knowledge (58%), knowledge of suppliers (58%), knowledge of the market (53%), company reputation (42%), and experience in running a business (42%).

In the second part of CATI, a question was asked to assess the attitude of respondents to the market situation in the face of the pandemic. A proactive attitude was found to prevail, as the respondents indicated such factors as: maintaining key clients (74%), maintaining financial liquidity (68%), acquiring new customers (63%) and maintaining relationships with suppliers (47%). Entrepreneurs also indicated that it was important to keep all jobs (47%) without the use of any form of state aid. As shown by the analysis of statistical data, only a few entrepreneurs managed to maintain employment at the pre-pandemic level. Short-term factors that only allowed survival included a reduction in employment (5%), reduction in wages and salaries (5%), or the acquisition of public aid in the form of the so-called "Anti-Crisis Shield" (5%).

The third set of questions concerned the sustainable management of enterprise resources. The responses were also classified into four groups corresponding to the above-mentioned types of resources, and the group "Strategy" was added. When asked about the actions taken by the enterprise to achieve Human Resource Management (HRM) in the time of the pandemic, the respondents primarily indicated: (a) shortening the working time of staff (47%), (b) maintaining contact with key customers (68%), (c) purchasing machinery and equipment (74%), and (d) activating marketing campaigns (47%).

The next questions concerned the company's stakeholders. The most important persons/organizations influencing the enterprise were indicated as follows:

(a) Among internal stakeholders: owner (47%), production employees (47%), sales employees (47%), all employees (37%), managers (32%), board members (26%), and family (16%);

(b)    Among external stakeholders: key customers (89%), raw material suppliers (84%), wholesale customers (47%), product suppliers (42%), service providers (42%), domestic competitors (42%), retail customers (37%), state authorities (37%), banks and other financial institutions (37%), and foreign competitors (37%).

In the next section, the respondents were asked to define with whom special relationships should be maintained during the pandemic; when answering, the importance of the relationship had to be indicated on a scale from 1 (not significant) to 5 (very high importance). Among intra-organizational relations, relations with sales employees (4.32), all employees (4.21), production employees (4.11), and owners (3.79) were of the highest importance. Significant external entities with which relations should be maintained included: key customers (4.79), suppliers of raw materials (4.47), wholesale clients (4.16), product suppliers (3.84), business partners (3.68), banks and financial institutions (3.42), and retail clients (3.32).

It was also interesting to find out who, according to the respondents, is held responsible for the survival/development of the company during the pandemic. Among internal stakeholders, the owner was indicated as the most important (4.37), and among external stakeholders, key customers were identified (4.63) (Table 1).

**Table 1.** Entities that the company depended on for survival/development during the pandemic.

| Stakeholders | Weighted Average Rating | Stakeholders | Weighted Average Rating |
|---|---|---|---|
| Internal Stakeholders | | External Stakeholders | |
| owner(s) | 4.37 | key customers | 4.63 |
| sales staff | 4.21 | raw material suppliers | 4.32 |
| production workers | 3.95 | wholesale clients | 4.05 |
| all employees | 3.79 | product suppliers | 3.74 |
| managers | 3.37 | business partners | 3.58 |
| marketing staff | 3.21 | retail clients | 3.26 |
| research and development employees | 3.00 | service providers | 3.26 |
| family | 2.95 | banks and other financial institutions | 3.16 |
| accountants | 2.89 | local competitors | 2.95 |
| warehouse workers | 2.79 | domestic competitors | 2.95 |
| board members | 2.74 | foreign competition | 2.89 |
| transport workers | 2.63 | logistics companies | 2.63 |
| IT department employees | 2.58 | IT companies | 2.63 |
| human resources workers | 2.47 | accounting firms | 2.53 |
| shareholders/stockholders | 2.42 | health service | 2.26 |
| colleagues, friends, acquaintances | 2.11 | local government authorities | 2.21 |
| administrative staff | 1.95 | industry authorities | 2.00 |
| | | ministries | 2.00 |
| | | state authorities | 1.95 |
| | | state offices | 1.95 |
| | | media | 1.68 |
| | | schools and universities | 1.63 |
| | | other | 1.53 |
| | | foundations, associations, clubs, unions | 1.42 |

The next examined issue was the question of relations. The respondents were asked to indicate the preferred type of relationship (formal, informal, social, business, official, from a position of strength) with a given group of stakeholders. For both internal and external stakeholders, formal relationships were most preferred (50% of responses in the group of internal stakeholders and 57% in the group of external stakeholders) (Tables 2 and 3).

**Table 2.** Preferred type of relationship when dealing with internal stakeholders.

| Internal Stakeholders | Type of Relationship * (Number of Indications) | | | | | |
|---|---|---|---|---|---|---|
| | F | I | S | B | O | ST |
| shareholders/stockholders | 12 | 2 | 3 | 8 | 0 | 0 |
| board members | 11 | 3 | 2 | 7 | 0 | 1 |
| owner(s) | 11 | 3 | 4 | 4 | 0 | 1 |
| managers | 9 | 5 | 6 | 11 | 0 | 0 |
| production workers | 10 | 6 | 4 | 2 | 0 | 1 |
| accountants | 10 | 2 | 0 | 6 | 0 | 0 |
| IT department employees | 11 | 2 | 2 | 6 | 0 | 0 |
| administrative staff | 10 | 3 | 2 | 5 | 1 | 0 |
| human resources workers | 11 | 3 | 2 | 6 | 0 | 0 |
| sales staff | 9 | 9 | 4 | 5 | 0 | 0 |
| warehouse workers | 12 | 5 | 1 | 3 | 0 | 1 |
| transport workers | 11 | 4 | 0 | 3 | 0 | 1 |
| marketing staff | 9 | 7 | 3 | 5 | 0 | 0 |
| research and development employees | 11 | 4 | 1 | 4 | 0 | 1 |
| all employees | 12 | 4 | 1 | 4 | 0 | 0 |
| family | 1 | 6 | 13 | 0 | 0 | 0 |
| colleagues, friends, acquaintances | 0 | 7 | 15 | 0 | 0 | 0 |
| other | 10 | 5 | 3 | 0 | 0 | 0 |
| Average number of indications | 9.44 | 4.44 | 3.67 | 4.39 | 0.06 | 0.33 |

* F—formal, I—informal, S—social, B—business, O—official, ST—from a position of strength.

**Table 3.** Preferred type of relationship when dealing with external stakeholders.

| External Stakeholders | Type of Relationship * (Number of Indications) | | | | | |
|---|---|---|---|---|---|---|
| | F | I | S | B | O | ST |
| raw material suppliers | 10 | 8 | 5 | 12 | 0 | 0 |
| product suppliers | 9 | 7 | 4 | 10 | 0 | 0 |
| service providers | 9 | 4 | 2 | 9 | 0 | 0 |
| IT companies | 10 | 1 | 2 | 8 | 0 | 0 |
| accounting firms | 10 | 2 | 1 | 8 | 0 | 0 |
| logistics companies | 10 | 1 | 1 | 8 | 0 | 0 |
| retail clients | 8 | 3 | 5 | 12 | 0 | 0 |
| wholesale clients | 8 | 4 | 6 | 11 | 0 | 0 |
| key customers | 9 | 6 | 7 | 12 | 0 | 0 |
| local government authorities | 11 | 0 | 0 | 2 | 10 | 0 |
| ministries | 10 | 0 | 0 | 1 | 10 | 0 |
| industry authorities | 11 | 1 | 1 | 3 | 7 | 0 |
| foundations, associations, clubs, unions | 11 | 3 | 2 | 0 | 4 | 0 |
| state authorities | 12 | 0 | 0 | 1 | 9 | 0 |
| media | 13 | 0 | 1 | 2 | 2 | 0 |
| banks and other financial institutions | 13 | 0 | 1 | 5 | 2 | 0 |
| local competitors | 11 | 6 | 5 | 7 | 0 | 0 |
| domestic competitors | 11 | 5 | 5 | 7 | 0 | 0 |
| foreign competition | 12 | 1 | 2 | 7 | 0 | 0 |
| schools and universities | 13 | 2 | 1 | 1 | 1 | 0 |
| state offices | 12 | 0 | 0 | 0 | 10 | 0 |
| business partners | 10 | 5 | 3 | 13 | 0 | 1 |
| health service | 14 | 2 | 0 | 0 | 4 | 0 |
| other | 12 | 4 | 0 | 0 | 0 | 0 |
| Average | 10.79 | 2.71 | 2.25 | 5.79 | 2.46 | 0.04 |

* F—formal, I—informal, S—social, B—business, O—official, ST—from a position of strength.

The information obtained during the interviews with the owners of enterprises allowed some phenomena identified in the quantitative research to be supplemented and/or explained. First, the aspect of the industry's success during the pandemic was supplemented with suggestions for a diversification strategy. Enterprises in the rubber products

industry had the opportunity to implement concentric diversification based on their production technology. This was spontaneously summarized by one of the respondents, who stated that "rubber products are everywhere". In the statements of the respondents, there were also concerns about the significant increase in the prices of machinery and equipment used in production, as well as the soaring shipping costs, as most of them are imported from China.

The interviews show that the SME sector in the rubber industry is significantly differentiated and divided into two basic groups:

(a)   Companies referred to in the industry as "garage companies", which use outdated machinery and have limited human resources. They try to catch up with leading manufacturers but, due to the lack of financial resources and a limited number of regular customers, are not able to fill competence and technological gaps; and

(b)   Companies that are constantly developing, investing in new technologies, machinery, and devices (injection molding machines, milling machines, lathes) and specialized measuring equipment (microscopes, sensors of production processes), training people, and adopting new systems specific to the industries for which they produce (e.g., the IATF 16,949 system dedicated to the automotive industry).

In addition, respondents highlighted the rising import prices from China impacting their manufacturing; e.g., as a result of the economic crisis caused by the COVID-19 pandemic, the price of stainless and black steel increased by 100%. This is confirmed by forecasts in the steel market. According to Hutnicza Izba Przemysłowo-Handlowa (Polish Steel Association), steel consumption in 2021 will return to the level of that in 2019 (in Poland, production decreased by 12% as a result of the pandemic, i.e., 7.89 million tonnes) [52]. Another problem in the field of sustainable resource management that respondents pointed out during the interviews was the drastic increase in the prices of shipping machinery and equipment from China (in so-called 40-foot containers), which increased from USD 4000 to USD 12,000, and the prices of silicone rubber, which increased from USD 2.5 per kilogram to EUR 4.5 per kilogram. Such drastic changes are additional arguments for the need to implement solutions in the analyzed SMEs that focus on concentric product diversification and the construction of a network of relations with stakeholders.

## 5. Discussion

In the interviews with representatives of the surveyed enterprises, it was confirmed that the entrepreneurs knew the consequences that could occur as a result of the crisis situation, which are described in the literature on the subject [13–15]. However, as a result of the COVID-19 pandemic, they had the opportunity to become aware of the real scale of the threat posed by an event that was long term, inherently abnormal, complex, and unstable, representing a threat to the strategic objectives, existence, or reputation of the organization. First, events that disrupt supply chains, which guarantee the continuity of the company's operations and maintain financial liquidity, are predominant [13,16–20]. It was noted that, so far, the recommendations suggested as an antidote to the crisis for micro and small entrepreneurs [13,21] are reactionary and short-lived (e.g., extending marketing and advertising campaigns, providing and promoting special offers and prices, lowering the prices of products and services, etc.). These are, of course, appropriate guidelines, especially for companies that did not conduct forecasting or preventive actions before the pandemic, did not include potential adverse scenarios in their planning activities, and did not implement a business continuity strategy.

The conducted research presented in this article confirms the appropriateness of the concept of stakeholder orientation and relationship building [24,38,39], especially with entities closely related to the value creation chain. This applies to both entities within the organization and those in its environment. In this context, it is important to focus on the need to redefine, or rather clarify, the concept of sustainable management. It is proposed that the perspective of balancing the interests of stakeholders [19,26,29–34,36,37] be disseminated while maintaining a narrow ecological focus [26–28].

With reference to the above conclusions, this article presents original definitions of sustainable organization, sustainable enterprise, and sustainable resource management, which, due to their common core, can be described as "sustainable survival".

Although the research sample ($n$ = 19) is not sufficient to draw conclusions and formulate recommendations, preliminary answers to the formulated research questions were obtained as a result of the conducted research. It was found that the most important resources of micro and small enterprises in the rubber products industry in Poland include: (a) human resources, particularly production and sales employees, (b) funds held by the enterprise (in a bank account or as cash), (c) raw materials for production and machinery and equipment for production, and (d) industry knowledge and knowledge of suppliers. It can be concluded that entrepreneurs from the analyzed sector pay special attention to securing production by maintaining existing supply chains. This is probably the reason why sustainable resource management, especially during a pandemic, is based on key customers, suppliers, and employees (related to production). There is an underlying logic in the respondents' answers: production and sales employees, key customers, and suppliers of raw materials were also indicated as key stakeholders. A similar distribution of responses also occurs in relation to entities with which the surveyed enterprises establish and maintain relationships, even during a pandemic.

Therefore, it can be concluded that the theses were verified on the basis of the results of the pilot study. In micro and small enterprises in the rubber products sector in Poland, sustainable resource management is related to the structure of the network of relations and the strength of connections in this network (relations/networking), as enterprises form a group of entities with a high level of loyalty, especially between the suppliers and buyers of raw materials. In a situation that threatens the survival of micro and small enterprises, the essence of sustainable resource management is concentric diversification with the goal of reducing the risk associated with dependence on a single recipient (enterprise, market, etc.).

Since the study focuses on small and medium-sized companies and the research sample is not sufficient, the formulated conclusions should become the basis for further in-depth research that can be conducted (a) in the same group of respondents, but using a representative research group, (b) in the same industry among a group of large enterprises, and (c) in a group of small and medium-sized enterprises (SMEs) from other industries. The following questions still remain to be investigated: (a) How did the threat caused by the COVID-19 pandemic affect the network of relationships and sustainable resource management in micro and small enterprises in the chemical industry in Poland? (b) For which resources is the link strongest between sustainable resource management and the structure of the relationship network and the strength of its connections? (c) Is there a relationship between the supply chain, the network of relationships, and sustainable resource management in micro and small enterprises in the chemical industry in Poland? In order to resolve the above-mentioned research questions, appropriate research will be undertaken. The following hypotheses were initially formulated: (H1) A situation that threatens the survival of micro and small enterprises in the chemical industry in Poland causes changes in the networks of relations, shifting them towards cooperation; (H2) A situation that threatens the survival of micro and small enterprises in the chemical industry in Poland causes changes in the networks of relations, shifting them towards secret connections.

## 6. Conclusions

The phenomenon defined in this article as "sustainable survival" is based on three pillars: (1) adoption of the perspective of key internal and external stakeholders by enterprises; (2) creation of a supply chain, the consistency of which results from the built relationships; (3) diversification of the risk associated with the possibility of disrupting the supply chain (blockage and/or loss of a chain link).

In connection with the above, two main paths related to the continuation of the studied topic emerge. In terms of considerations and theoretical research, it is worth

(1) ensuring a consistent understanding of the concept of sustainable management so that it is not dominated by a particular aspect (e.g., ecological), which, by definition, contradicts the idea of sustainability; (2) conducting further, extended, and in-depth research related to the creation of a supply chain, integrated by relationships and diversified according to the specificity of the sector and the size of the enterprise; (3) disseminating and developing stakeholder relationship management concepts in the context of sustainable management/development.

Since the research presented in this article concerns micro and small enterprises from a specific industry and country, the formulated conclusions apply mainly to this group. It should also be emphasized that, in this case, the intended recipients of the message and the main decision-makers are the owners of these enterprises. Therefore, it becomes apparent that the basic activity related to the transfer of knowledge to these entities should consist in reaching business owners with information. Among these decision-makers, it is worth promoting the ideas of cooperation, loyalty, and collaboration. In the sector of medium and large enterprises, the above-mentioned solutions take formalized forms of coopetition, strategic alliances, capital groups, etc. However, on a smaller scale, they are still uncoordinated, informal, unexplored, and unnamed.

In economies such as Poland, where micro and small enterprises play a key role, it is also worth considering system solutions that support the cooperation of micro and small enterprises. The COVID-19 pandemic has shown that solutions based on financial support for enterprises from central funds are effective but capital intensive and insufficient in the long term. As shown in this research, this help may even be unnecessary when companies strategically build the foundations for their stability, i.e., create relationships with stakeholders.

**Author Contributions:** Conceptualization, A.S., E.I.T. and K.C.; methodology, A.S., E.I.T. and K.C.; software, A.S. and E.I.T.; validation, A.S., E.I.T. and K.C.; formal analysis, K.C.; investigation, A.S.; resources, A.S., E.I.T. and K.C.; data curation, A.S., E.I.T. and K.C.; writing—original draft, A.S., E.I.T. and K.C.; writing—review and editing, E.I.T. visualization, A.S., E.I.T. and K.C.; supervision, A.S., E.I.T. and K.C.; project administration, A.S., E.I.T. and K.C. All authors have read and agreed to the published version of the manuscript.

**Funding:** This research was funded by: GENERAL TADEUSZ KOŚCIUSZKO MILITARY UNIVERSITY OF LAND FORCES, grant number 168/WZA/83/DzS (67%), and private funds of Eleftherios I. Thalassinos (33%). The APC was funded by 1/GENERAL TADEUSZ KOŚCIUSZKO MILITARY UNIVERSITY OF LAND FORCES (67%) and private funds of Eleftherios I. Thalassinos (33%).

**Institutional Review Board Statement:** Not applicable.

**Informed Consent Statement:** Not applicable.

**Data Availability Statement:** The data presented in this study are available on request from the corresponding author. The data are not publicly available because website or remote access was not created.

**Acknowledgments:** The authors would like to thank Marta Wincewicz-Bosy for enabling the team to carry out scientific and research work.

**Conflicts of Interest:** The authors declare no conflict of interest. The funders had no role in the design of the study; in the collection, analyses, or interpretation of data; in the writing of the manuscript or in the decision to publish the results.

**Appendix A**

The questionnaire used in the CATI study.

| Period of Operation on the Market (years) | | | | | Number of Employees | | | | | | Status | | |
|---|---|---|---|---|---|---|---|---|---|---|---|---|---|
| More than 20 | 19–15 | 14–10 | 9–3 | Less than 3 | 1 | 2–5 | 6–10 | 11–20 | 21–30 | 31–50 | Owner | Manager | Employee |
| | | | | | | | | | | | | | |

1. Which resources are most important to your company in the current situation?

| | | |
|---|---|---|
| **People** | production workers | |
| | sales staff | |
| | marketing staff | |
| | accounting and finance staff | |
| | managers | |
| | other workers | |
| **Money** | cash | |
| | funds on the company's account | |
| | funds from the Shield | |
| | funds from loans and credits | |
| | receivables | |
| **Property resources** | good localization | |
| | real estate | |
| | machines and devices | |
| | means of transport | |
| | raw materials | |
| | another | |
| **Information resources** | industry knowledge | |
| | experience in running a business | |
| | reputation | |
| | knowledge of the market | |
| | knowledge of suppliers | |

2. What matters in the current economic climate?

| | |
|---|---|
| Survive in the market | |
| Get as much as you can | |
| Strengthen your business | |
| Develop your business | |
| Enter e-business | |
| Keep all jobs | |
| Reduce employment | |
| Reduce payroll costs | |
| Obtain funds from the Shield | |
| Obtain a preferential credit/loan | |
| Maintain financial liquidity | |
| Maintain relationships with suppliers | |
| Acquire new suppliers | |
| Retain key customers | |
| Acquire new customers | |
| Other | |

3.  What actions are you taking to manage your resources in a sustainable way?

| | | |
|---|---|---|
| People | reduction in wages | |
| | increasing wages | |
| | layoffs | |
| | recruitment | |
| | free holidays | |
| | mandatory holidays | |
| | increasing the scope of responsibilities | |
| | replacement of permanent employees with part-time temporary employees | |
| | reduction in staff working time | |
| | other | |
| Strategy | maintaining contact with key customers | |
| | entering into cooperation | |
| | the use of new technologies to reduce operating costs | |
| | starting e-commerce activity | |
| | acquiring new customers | |
| | acquiring new sales markets | |
| | entering new segments of the customer market | |
| | introduction of new products and services | |
| | special offers and prices | |
| | high discounts | |
| | lowering prices | |
| Money | deferment of own payments | |
| | using the Shield | |
| | obtaining credits and loans | |
| | extending the repayment date of receivables | |
| | other | |
| Property resources | renting own space to other entities | |
| | purchase of machinery and equipment | |
| | sale of machinery and equipment | |
| | purchase of means of transport | |
| | sale of means of transport | |
| | other | |
| Information resources | participation in training | |
| | using business consulting | |
| | extending marketing and advertising campaigns | |
| | other | |

4.  Who are the most important people/organizations influencing your company?

| | | |
|---|---|---|
| Internal stakeholders | shareholders/stockholders | |
| | board members | |
| | owners | |
| | managers | |
| | production workers | |
| | accountants | |
| | IT department employees | |
| | administrative staff | |
| | staff workers | |
| | sales staff | |
| | warehouse workers | |
| | transport workers | |
| | marketing staff | |
| | research and development employees | |
| | all employees | |
| | family | |
| | colleagues, friends, acquaintances | |
| | other | |
| External stakeholders | raw material suppliers | |
| | product suppliers | |
| | service providers | |
| | IT companies | |
| | accounting firms | |
| | logistics companies | |
| | retail customers | |
| | wholesale customers | |
| | key customers | |
| | local government authorities | |
| | ministries | |
| | industry authorities | |
| | foundations, associations, clubs, associations | |
| | state authorities | |
| | media | |
| | banks and other financial institutions | |
| | local competitors | |
| | domestic competitors | |
| | foreign competition | |
| | schools and colleges | |
| | state offices | |
| | business partners | |
| | health service | |
| | other | |

5. With whom should you maintain special relationships during a pandemic? Rate the importance of the relationship on a scale from 0 to 5.

| | | |
|---|---|---|
| Internal stakeholders | shareholders/stockholders | |
| | board members | |
| | owners | |
| | managers | |
| | production workers | |
| | accountants | |
| | IT department employees | |
| | administrative staff | |
| | staff workers | |
| | sales staff | |
| | warehouse workers | |
| | transport workers | |
| | marketing staff | |
| | research and development employees | |
| | all employees | |
| | family | |
| | colleagues, friends, acquaintances | |
| | others | |
| External stakeholders | raw material suppliers | |
| | product suppliers | |
| | service providers | |
| | IT companies | |
| | accounting firms | |
| | logistics companies | |
| | retail customers | |
| | wholesale customers | |
| | key customers | |
| | local government authorities | |
| | ministries | |
| | industry authorities | |
| | foundations, associations, clubs, associations | |
| | state authorities | |
| | media | |
| | banks and other financial institutions | |
| | local competitors | |
| | domestic competitors | |
| | foreign competition | |
| | schools and colleges | |
| | state offices | |
| | business partners | |
| | health service | |
| | other | |

6. Who, in a pandemic, could depend on the survival/development of your business? Rate the strength of the impact on a scale from 0 to 5.

| | | |
|---|---|---|
| Internal stakeholders | shareholders/stockholders | |
| | board members | |
| | owners | |
| | managers | |
| | production workers | |
| | accountants | |
| | IT department employees | |
| | administrative staff | |
| | staff workers | |
| | sales staff | |
| | warehouse workers | |
| | transport workers | |
| | marketing staff | |
| | research and development employees | |
| | all employees | |
| | family | |
| | colleagues, friends, acquaintances | |
| | other | |
| External stakeholders | raw material suppliers | |
| | product suppliers | |
| | service providers | |
| | IT companies | |
| | accounting firms | |
| | logistics companies | |
| | retail customers | |
| | wholesale customers | |
| | key customers | |
| | local government authorities | |
| | ministries | |
| | industry authorities | |
| | foundations, associations, clubs, associations | |
| | state authorities | |
| | media | |
| | banks and other financial institutions | |
| | local competitors | |
| | domestic competitors | |
| | foreign competition | |
| | schools and colleges | |
| | state offices | |
| | business partners | |
| | health service | |
| | other | |

7. What kind of relationship do you prefer when dealing with individual people/organizations?
   F—formal, N—informal, T—social, B—business, U—official, ST—position of strength.

| | | |
|---|---|---|
| Internal stakeholders | shareholders/stockholders | |
| | board members | |
| | owners | |
| | managers | |
| | production workers | |
| | accountants | |
| | IT department employees | |
| | administrative staff | |
| | staff workers | |
| | sales staff | |
| | warehouse workers | |
| | transport workers | |
| | marketing staff | |
| | research and development employees | |
| | all employees | |
| | family | |
| | colleagues, friends, acquaintances | |
| | other | |
| External stakeholders | raw material suppliers | |
| | product suppliers | |
| | service providers | |
| | IT companies | |
| | accounting firms | |
| | logistics companies | |
| | retail customers | |
| | wholesale customers | |
| | key customers | |
| | local government authorities | |
| | ministries | |
| | industry authorities | |
| | foundations, associations, clubs, associations | |
| | state authorities | |
| | media | |
| | banks and other financial institutions | |
| | local competitors | |
| | domestic competitors | |
| | foreign competition | |
| | schools and colleges | |
| | state offices | |
| | business partners | |
| | health service | |
| | other | |

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
