# Peer review of "Sustainable Survival: Resource Management Strategy in Micro and Small Enterprises in the Rubber Products Market in Poland during the COVID-19 Pandemic"

_resources, doi:10.3390/resources10080085_

Round 1

Reviewer 1 Report

The manuscript makes an attempt to assess the resource management strategy of micro and small enterprises of rubber products in Poland. Nevertheless, its current version is too simplistic and descriptive, without presenting an innovative contribution in the scope of the relevant framework of sustainable performance of micro, small and medium-sized firms. In my view, it needs further refinement in terms of the paper's position and value-added, literature review, and case studies development. 

Reviewer 2 Report

The article is interesting, and the researched problem has scientific potential. However, some problems need to be solved:

  1. The abstract must be restructured in such a way as to provide the information required in the template requirements ("For research articles, abstracts should give a pertinent overview of the work. We strongly encourage authors to use the following style of structured abstracts, but without headings: (1) Background: Place the question addressed in a broad context and highlight the purpose of the study; (2) Methods: Describe briefly the main methods or treatments applied; (3) Results: Summarize the article's main findings; and (4) Conclusions: Indicate the main conclusions or interpretations."). Research questions are not usually listed in the abstract.

  1. The structure of the article is cumbersome and needs to be reorganized. The introduction is too rich in information. As the template for sustainability articles says: "The introduction should briefly place the study in a broad context and highlight why it is important. It should define the purpose of the work and its significance". Authors must "briefly mention the main aim of the work and highlight the main conclusions." The introduction needs to be restructured and synthesized

  1. Literature review studies sustainable development to sustainable resource management, in general, and to a very small extent studies micro and small enterprises. Literature review needs to study more resource management strategies in micro and small enterprises.

  1. The authors must provide (eventually in the appendix) the structure of the questionnaire used.

  1. The hypotheses are not clearly formulated (they are briefly mentioned at the end of the article).

  1. Data processing is performed using descriptive statistics. The article would gain value if complex statistical methods were used to establish the relationships between variables (SEM, MANOVA, multiple regressions, etc.)

  1. The discussion section should be built in the context of dialogue with researchers in the literature review.

  1. In my opinion, a section of conclusions that includes theoretical and managerial implications, research limitations, and future research directions would be helpful.

The article presents some scientific value and can be published after a careful review of the reported issues.

Reviewer 3 Report

This article aims to analyze the phenomenon of sustainable resource management in micro and small companies, in the rubber products industry in Poland, during the COVID-19 pandemic.

The introduction effectively presents the problem under study and the framing of the problem is well structured. The methodology is suited to the way the authors intend to deal with the topic. However, there are some parts of the work that can be improved. Below I present my suggestions.

Abstract: Do not present the hypotheses under study, but the problem under study. Authors should follow the following structure: background, brief description of the methods used, summarize the results and present the main priorities. Additional efforts should be made to identify the contributions of this study to theory and practice.

Theoretical framework: The framework should be more in line with the research questions the authors identify in the abstract.

Results: It is necessary to relate the results obtained in this study with the evidence obtained in previous studies (in order to highlight the value that this study assumes in relation to the existing literature);

It is highly recommended to delve into the limitations and, based on them, in future lines of research. The authors devote very limited space to this aspect and greater effort must be made.

I hope the authors can consider these suggestions in order to qualitatively improve their article.

Round 2

Reviewer 1 Report

Congratulations on the revised version of the manuscript.

Author Response

Dear Reviewer,

Thank you very much for your constructive criticism, inspiring and valuable comments.

Yours sincerely
Authors

Reviewer 2 Report

The paper can be published in present form.

Author Response

(The authors gave the same response as above.)

Reviewer 3 Report

The authors made a number of significant improvements to the article, which thus deserves a review. After a careful analysis of this second version of the article, I congratulate the majority of the improvements introduced, as some parts of the work still need a major revision.
Below I leave my comments and suggestions for improvement.

1. In relation to the interviews, much more information is necessary. In particular, who performs these interviews? What procedure is used? When are they done? By what are they carried out?

2. Tables must be renumbered. There is no table 1.

3. In line 324 of the second version of the article, the authors mention figure 3, but I do not detect figures in this version.

4. On line 329, what does "(ZZZ)" mean?

5. The conclusions presented need to be reformulated:
- Since the study focuses on small and medium-sized companies, there are no data that allow us to draw conclusions about medium and large companies;
- There is no alignment between the starting questions with which the authors started the study and the conclusions they reach;
- It is possible to detect contributions from the study to practice, which need further substantiation;
- Contributions to theory, study limitations and suggestions or clues for future studies are still lacking.

6. Final reflection for authors: Authors should bear in mind the need to detect an alignment of ideas that facilitates reading and makes the article structurally more organized. The questions that start the study must be answered at the end. Throughout the article, the path taken to reach the conclusions must be made clear.

Author Response

Dear Reviewer,
Thank you very much for your constructive criticism, inspiring and valuable comments. Below are the answers to the individual points of the review.

Yours sincerely
Authors

  1. In relation to the interviews, much more information is necessary. In particular, who performs these interviews? What procedure is used? When are they done? By what are they carried out?

Added explanation and schematic of the IDI procedure (lines 277-278 and 288 figure 1)

2. Tables must be renumbered. There is no table 1.

Changed

3. In line 324 of the second version of the article, the authors mention figure 3, but I do not detect figures in this version.

Changed

4. On line 329, what does "(ZZZ)" mean?

Changed

5. The conclusions presented need to be reformulated:
- Since the study focuses on small and medium-sized companies, there are no data that allow us to draw conclusions about medium and large companies;

Changed - see lines 495-513 and from 523

- There is no alignment between the starting questions with which the authors started the study and the conclusions they reach;

- It is possible to detect contributions from the study to practice, which need further substantiation;
- Contributions to theory, study limitations and suggestions or clues for future studies are still lacking.

Changed - see lines 495-513 and form 523

6. Final reflection for authors: Authors should bear in mind the need to detect an alignment of ideas that facilitates reading and makes the article structurally more organized. The questions that start the study must be answered at the end. Throughout the article, the path taken to reach the conclusions must be made clear.

Changed - see lines from 471

Round 3

Reviewer 3 Report

Congratulations on the changes made. Good luck for future work.